# Stuck in a lockdown: Dreams, bad dreams, nightmares, and their relationship to stress, depression and anxiety during the COVID-19 pandemic

Elizaveta Solomonova[1,2]*, Claudia Picard-Deland[3,4], Iris L. Rapoport[5], Marie-Hélène Pennestri[6,7], Mysa Saad[8], Tetyana Kendzerska[9], Samuel Paul Louis Veissiere[2], Roger Godbout[7,11], Jodi D. Edwards[10,12], Lena Quilty[13], Rebecca Robillard[8,14]

1 Neurophilosophy Lab, McGill University, Montreal, Canada, 2 Division of Social and Transcultural Psychiatry, Department of Psychiatry, Culture, Mind and Brain research group, McGill University, Montreal, Canada, 3 Dream and Nightmare Laboratory, Center for Advanced Research in Sleep Medicine, Montreal, Canada, 4 Department of Neuroscience, University of Montreal, Montreal, Canada, 5 Department of Philosophy, McGill University, Montreal, Canada, 6 Department of Educational and Counselling Psychology, McGill University, Montreal, Canada, 7 Sleep Laboratory and Clinic, Hôpital en Santé Mentale Rivière-des-Prairies, CIUSSS du Nord-de-l'Île-de-Montréal, Montreal, Canada, 8 The Royal's Institute of Mental Health Research, Ottawa, Canada, 9 Department of Medicine, The Ottawa Hospital Research Institute, University of Ottawa, Ottawa, Canada, 10 University of Ottawa Heart Institute, Ottawa, Canada, 11 Department of Psychiatry, University of Montreal, Montreal, Canada, 12 Department of Psychiatry, Centre for Addiction and Mental Health, University of Toronto, Toronto, Canada, 13 School of Epidemiology and Public Health, University of Ottawa, Ottawa, Canada, 14 School of Psychology, University of Ottawa, Ottawa, Canada

* elizaveta.solomonova@mail.mcgill.ca

## Abstract

### Background

An upsurge in dream and nightmare frequency has been noted since the beginning of the COVID-19 pandemic and research shows increases in levels of stress, depression and anxiety during this time. Growing evidence suggests that dream content has a bi-directional relationship with psychopathology, and that dreams react to new, personally significant and emotional experiences. The first lockdown experience was an acute event, characterized by a combination of several unprecedent factors (new pandemic, threat of disease, global uncertainty, the experience of social isolation and exposure to stressful information) that resulted in a large-scale disruption of life routines. This study aimed at investigating changes in dream, bad dream and nightmare recall; most prevalent dream themes; and the relationship between dreams, bad dreams, nightmares and symptoms of stress, depression and anxiety during the first COVID-19 lockdown (April-May 2020) through a national online survey.

### Methods

968 participants completed an online survey. Dream themes were measured using the Typical Dreams Questionnaire; stress levels were measured by the Cohen's Perceived Stress Scale; symptoms of anxiety were assessed by Generalized Anxiety Disorder (GAD-7)

**Data Availability Statement:** Data cannot be made publicly available due to restrictions imposed by

the Clinical Trials Ontario-Qualified Research Ethics Board concerning participant confidentiality. Proposals to access data from this study can be submitted to the corresponding author or to Caitlin Higginson (contact via caitlin.higginson@theroyal.ca) and the data may thus be made available upon data sharing agreement.

**Funding:** The funders had no role in study design, data collection and analysis, decision to publish, or preparation of the manuscript.

**Competing interests:** The authors have declared that no competing interests exist.

scale; and symptoms of depression were assessed using the Quick Inventory of Depressive Symptomatology.

## Results

34% (328) of participants reported increased dream recall during the lockdown. The most common dream themes were centered around the topics of 1) inefficacy (e.g., trying again and again, arriving late), 2) human threat (e.g., being chased, attacked); 3) death; and 4) pandemic imagery (e.g., being separated from loved ones, being sick). Dream, bad dream and nightmare frequency was highest in individuals with moderate to severe stress levels. Frequency of bad dreams, nightmares, and dreams about the pandemic, inefficacy, and death were associated with higher levels of stress, as well as with greater symptoms of depression and anxiety.

## Conclusions

Results support theories of dream formation, environmental susceptibility and stress reactivity. Dream content during the lockdown broadly reflected existential concerns and was associated with increased symptoms of mental health indices.

## 1. Introduction

Since the declaration of the global COVID-19 pandemic in March 2020 by the World Health Organization and since measures of social isolation have been in place in many countries, including Canada, dreams have been a hot topic of conversation. A number of popular outlets reported an increase in dream reports and specifically in bizarre, threatening dreams [1–5], and social media became saturated with reports of new dream experiences, all related to elements of the COVID-19 pandemic–a phenomenon that Nielsen called "a dream surge" [6]. In line with renewed interest in dreams during the pandemic, and following early anecdotal evidence for intensified dreaming, growing research provides evidence for specific changes in dream content during the acute stage of the health crisis. Dreams, currently defined as any cognitive activity that happens during sleep which has a subjective component that is recalled while awake [7], are known to react to one's most pressing concerns, of both personal and collective nature, and are thought to play a role in the integration of lived experiences, in extracting the gist of new information and in emotion regulation [8]. The current pandemic offers a unique opportunity to investigate changes to our oneiric lives brought about by shared concerns during these times of dealing with illness or threat of illness and in the context of an unprecedented degree of change in socio-economic aspects of life. In this paper, we present results of an ongoing national online survey focused on multiple facets of mental health, social and economic impacts of COVID-19 and including validated sleep and dream questionnaires. Our project had three goals: 1) to qualify changes in dream, bad dream and nightmare recall; 2) to identify the most prevalent dream themes; and 3) to assess the relationship between dreams, stress, and symptoms of anxiety and depression during the first lockdown phase of COVID-19.

### 1.1. COVID-19, sleep and mental health

An emerging body of research shows the many ways in which the experience of the pandemic, and in particular of social isolation, is affecting sleep specifically and mental health more

broadly, notably with documented increases in symptoms of depression and anxiety [9–14]. Mounting evidence shows widespread changes in sleep patterns, including increased sleep duration, later bedtimes and later awakenings, as well as a reduction in social jet lag (a form of circadian misalignment between one's endogenous best rhythm and the demands of the social world, such as early work or school start) [15–20]. Nevertheless, in a number of studies, increased sleep time and opportunities for sleep schedules more aligned with one's chronotype did not preclude reports of poor sleep quality, likely due to increased levels of stress associated with the pandemic, lockdown, and personal challenges [17, 19, 21, 22]. Unsurprisingly, then, increased rates of insomnia were also reported by multiple teams [23–25]. Using data from the larger cohort of the current study, our group has previously reported an important increase in stress levels during the pandemic, a relationship between increased stress and pre-existing mental illness [9, 10], as well as changes in patterns of sleep duration (increase vs. decrease) and timing (delay) during the pandemic [20]. The role of disordered sleep in exacerbating mental disorders, in particular depression and anxiety, is well documented [26]. During confinement, poor sleep was similarly associated with higher levels of psychotic-like experiences, rumination and somatic symptoms [27]. This bi-directional relationship between sleep and mental health may also be evidenced by dysphoric or intensified dreaming.

## 1.2. Dream content: Mental health and the relationship to lived experiences

Dream content is often seen as a proxy for understanding individual experience of health and distress: dreams are known to react to new experiences and life changes. Bad dreams and nightmares tend to increase during periods of high stress [28–30], and are often exacerbated by, and potentially play a role in, flare-ups of mental health issues, including depression [31] and anxiety [32] In general, many current theories of dream formation and of dream function converge on the following main ideas: it is suggested that dream content 1) reflects elements of daily lived experience [33–35]; 2) extracts its meaning by creatively integrating new information into existing autobiographical memory networks [36–40]; and 3) preferentially responds to current concerns and to emotionally charged events with high personal significance [34, 41].

In addition to individual experiences, collective trauma can affect the content and intensity of dreams in the general population. For example, studies tracking dreams before and after the 9/11 attacks observed an increase in nightmare frequency in male respondents [42], an increase in the intensity of dream imagery [43] and clear incorporations of a variety of 9/11-related topics, such as planes crashing into buildings, being hijacked and bombs [44]. Similarly, studies tackling dream content in prisoners of wars and Holocaust survivors [45, 46] revealed a profound intrusion of war-related themes in dreams, as well as a long-lasting increase in dream negative emotions and threatening events. Lastly, a study on WWII Veterans [47] showed more frequent nightmares and symptoms of insomnia, depression and anxiety that persisted even decades after the war.

The current pandemic, and in particular the experience of a lockdown, presents a set of new challenges to our psychological and social health. These include novel experiences which have a strong emotional component and of which we are, individually and collectively, attempting to make sense. It is therefore unsurprising that dreams should change during these times, as a process of making sense of and of adapting to these novel aspects of life. Thus, this historical period represents a unique and naturalistic opportunity to explore current theories of dream formation, including frameworks of environmental susceptibility and stress reactivity.

### 1.3. Typical dream themes

Previous work has demonstrated a number of common themes in dream content. The Typical Dreams Questionnaire (TDQ), first used in a sample of Canadian University students [48], investigates the relative prevalence of different dream themes. This instrument has been validated in a German [49] and in a Chinese [50] samples. Results of the three studies converge on the possibility that dream themes are relatively stable across the individual life span and are relatively similar across cultures, potentially representing shared concerns, fears, and aspirations common to members of the globalized world. The most common themes in the dreams of the Canadian sample, specifically, were: being chased, sexual experiences, falling, school/studying and arriving too late [48]. The current pandemic presents a new and unique context for the study of personal and collective experiences. This novel context, characterized by such concerns and experiences as the threat and experience of contagious disease, social isolation, personal and economic uncertainty, lockdowns, and others, quite unprecedentedly, is shared across most countries. We hypothesize that new dream themes will emerge as most prevalent during the lockdown in response to these pandemic-specific challenges.

### 1.4. Dreams during COVID-19

An increasing amount of studies report changes in dream content during the current pandemic. An upsurge of dream [51, 52] and nightmare [53–57] frequencies has been observed during lockdown, especially for the female sex [54, 58, 59], for individuals with sleep disturbances, anxiety or depressive symptoms [51, 58] and for individuals with COVID-related increases in stress levels [57]. An increase in nightmares without COVID-19 imagery was also observed in Canadian patients previously diagnosed with primary PTSD [60]. Furthermore, the presence of COVID-19 related nightmares has been associated with symptoms of generalized anxiety disorder in healthcare providers in Colombia [61]. Similarly, the presence of COVID-19 related dreams in a large Spanish sample was associated with many factors, including having infected family or friends, reading news about coronavirus, and having higher depression, anxiety or stress scores [62]. With regards to specific features of dream content, Brazilian [63], U.S. [64], Italian [51, 58, 65], Canadian [54] and Chinese [66] teams all reported increases in negative dreams during the pandemic, including increase in themes of anxiety, sadness, aggressive interactions and preoccupations with health. Similarly, a U.S. survey revealed that 15% of the participants' dreams shifted towards more negative dream emotions [52]. Using automatic text algorithms on dream content during lockdown, Pesonen et al. [57] found that a majority of distressing dreams were pandemic-specific, with themes such as failures in social distancing, coronavirus contagion, dystopia, and apocalypse. These dreams were more accentuated in individual with higher perceived stress and, while the imagery was pandemic-specific, they were nonetheless similar to typical idiopathic dream themes (e.g. failure, death). Lastly, in a sample of Canadian university students [67], dreams during the early weeks of the COVID-19 pandemic were characterized by increased virus imagery as well as by animal imagery and location changes.

### 1.5. Study objectives and hypotheses

The objectives of the current study were to characterize dream, bad dream and nightmare recall during the first lockdown phase of the COVID-19 pandemic; to identify the most prevalent dream themes; and to investigate the relationship between dream and psychological variables, specifically: the degree of concern with relation to different facets of the pandemic, levels of stress, as well as symptoms of anxiety and depression. We hypothesized that: 1) dream, bad dream and nightmare recall would increase; 2) that pandemic- and threat-related imagery

would be frequent and reflect ongoing concerns; 3) that higher dream, bad dream and night-mare recall, as well as the most prevalent dream topics during the pandemic, would be associated with higher levels of stress, with more concerns over COVID-19 and greater symptoms of depression and anxiety; and 4) that, keeping in mind the bi-directional nature of the relationship between dysphoric dreaming and psychopathology, bad dream and nightmare frequency as well as pandemic- and threat- related imagery would in turn predict symptoms of stress, depression and anxiety.

## 2. Materials and methods

### 2.1. Survey

We present data from the ongoing longitudinal national "How are you coping" web-based survey. The survey aims at characterizing the many facets of the COVID-19 pandemic's impact on different aspects of mental health. The full methodology of the survey can be found elsewhere [9]. The study was approved by the Clinical Trials Ontario-Qualified Research Ethics Board (Protocol #2131). Electronically informed consent was obtained from participants prior to starting the survey. The complete survey contained a main part assessing psychiatric, social, economic and health impacts of the pandemic, and an optional component. Dream themes were included in the optional component.

### 2.2. Participants

A total of 968 participants (12–92 years old, Mean age = 52.5±17.2; Female = 710, Male = 258) completed the optional component of the survey between April 3 and May 15 2020. Most participants ($n$ = 940) were located in Canada during the outbreak, 12 in the United States, 4 in France, 3 in the United Kingdom, 1 in Australia and 1 in New Zealand, and 7 declined to provide location. 135 (14%) of the participants reported working in essential services (not closed during the lockdown), of which 13 worked in a hospital, 6 in a grocery store, 1 in public transportation, and 36 in other essential services.

### 2.3. Measures

**Dream themes** were assessed using a modified version of the Typical Dreams Questionnaire (TDQ) [48]. In addition to the 56 themes from the original questionnaire, we added 4 items (germs or being contaminated, being in a hospital, being sick, and being separated from a loved one), representative of some of the major concerns during the pandemic. While the original TDQ focuses on the lifetime prevalence of each theme, we asked participants to rate how often they had dreamt of each theme during the past week. Overall dream theme prevalence (% of people endorsing each item at least once), and the frequency of experience of each dream theme (0 = "never", 1 = "once", 2 = "2–3 times", 3 = "4–10 times", 4 = "11+ times") were calculated. Individual dream themes were then combined into 7 larger topics: Pandemic, Natural Threat, Human Threat, Inefficacy, Paralysis, Death, and Paranormal imagery. Prevalence for each topic was calculated as the percentage of people that dreamt at least once of an item included in that topic. The list of specific items that were used in each topics is presented in Table 1.

**Dream, bad dream and nightmare recall** were assessed by asking participants 1) the average weekly number of dreams, bad dreams or nightmares they recalled in the past year preceding the lockdown, and 2) in the past 7 days.

**Pandemic-related concerns** were assessed using the following questions: participants were asked to rate how concerned they currently were about: 1) lacking food, 2) public services

**Table 1. Dream topics and dream themes.**

| Dream topics | TDQ items |
|---|---|
| **Pandemic**[a] | *germs or being contaminated (#57)* |
| | *being sick (#58)* |
| | *being in hospital (#59)* |
| | *being separated from a loved one (#60)* |
| **Natural Threat** | *snakes (#9)* |
| | *flood or tidal waves (#21)* |
| | *tornadoes or strong winds (#22)* |
| | *earthquakes (#23)* |
| | *insects or spiders (#24)* |
| | *fire (#34)* |
| | *wild, violent beasts (#40)* |
| **Human Threat** | *being chased or pursued, but not physically injured (#1)* |
| | *being physically attacked (beaten, stabbed, raped, etc.) (#2)* |
| | *vividly sensing a presence in the room (#29)* |
| **Inefficacy** | *trying again and again to do something (#3)* |
| | *arriving too late, e.g., missing a train (#6)* |
| | *being locked up (#8)* |
| | *losing control of a vehicle (#33)* |
| | *failing an examination (#38)* |
| **Paralysis** | *being frozen with fright (#4)* |
| | *being tied, unable to move (#15)* |
| | *being smothered, unable to breathe (#39)* |
| | *being half awake and paralyzed in bed (#44)* |
| **Death** | *being killed (#27)* |
| | *seeing yourself as dead (#28)* |
| | *a person now dead as alive (#35)* |
| | *a person now alive as dead (#36)* |
| | *killing someone (#42)* |
| **Paranormal** | *having superior knowledge or mental ability (#16)* |
| | *having magical powers (#20)* |
| | *seeing a UFO (#46)* |
| | *seeing extra-terrestrials (#47)* |
| | *traveling to another planet or visiting a different part of the universe (#48)* |
| | *being an animal (#49)* |
| | *seeing an angel (#51)* |
| | *encountering God in some form (#52)* |
| | *encountering a kind of evil force or demon (#56)* |

[a] Items not included in the original version of TDQ-56.

shutting down; 3) schools shutting down or staying closed for an extended period; 4) their children or relatives not coping well with the situation; and 5) not being able to access medications or medical services. Each type of concern was rated on a sliding scale from 0 (not concerned at all) to 100 (very concerned).

Participants were asked to rate their stress, anxiety and depression levels for the last 7 days. **Stress levels** were assessed using Cohen's Perceived Stress Scale (PSS) [68]. Participants were then separated into low (0 = 13) moderate (14–25) or severe (>27) stress groups based on the

PSS scores. **Anxiety symptoms** were evaluated using the Generalized Anxiety Disorder (GAD-7) 7-item scale [69], and **depressive symptoms** were investigated using the Quick Inventory of Depressing Symptomatology (QIDS) scale [70]. **Sleep duration** (in hours per day,) was assessed as part of the Pittsburgh Sleep Quality Index (PSQI) questionnaire [71] adjusted to reflect two periods: 1) the last month before the outbreak, and 2) the last 7 days.

## 2.4. Statistical analyses

Wilcoxon signed-rank related samples tests were used to assess changes in dream recall, bad dream and nightmare frequency between pre-COVID-19 and the last 7 days. Descriptive statistics were used to assess the prevalence of each individual dream theme and of broader dream topics. Spearman correlations were performed to investigate the relationship between the frequency of the most prevalent dream themes, levels of concern for specific aspects of the pandemic, levels of stress, and symptoms of anxiety and depression.

A MANCOVA with dream, bad dream and nightmare recall as dependent measures and with stress levels (low, medium, high) as independent factors, and age and sex as covariates, was performed to test whether dream variables were associated with stress levels. Follow-up separate univariate ANOVAs were used to investigate potential differences in dream, bad dream and nightmare recall across stress levels (low, moderate, severe). A MANCOVA with most frequent dream topics as dependent variables, stress levels (low, moderate, severe) as independent factors and dream recall, age and sex as covariates was performed to test the relationship between stress group and dream topics. Separate follow-up univariate ANCOVAs with stress levels (low, moderate and severe) as independent factors, dream recall as a covariate and frequency of four most prevalent dream topics as dependent factors were performed to assess the relationship between stress and specific dream topics.

To investigate the potential predictor role for dream variables on mental health outcomes, we performed separate linear regressions with dream recall, bad dream and nightmare frequency as predictors; age, sex, income (above/below CAD 40,000 per year), employment status (employed/unemployed/retired) and antidepressant use as covariates; and stress, depression and anxiety levels as dependent variables.

## 3. Results

### 3.1. Dream recall and dream themes

**3.1.1. Dream, bad dream and nightmare recall.** A Wilcoxon signed-rank test revealed a weak decrease in dream recall from the year before the pandemic to the last 7 days ($z = 4.491$, $p < .001$). Mean dream recall was 3.26±4.97 dreams per week in the year pre-pandemic, and 2.86±2.02 dreams per week in the last 7 days. (Median before pandemic and in the last 7 days = 2 dreams per week)

443 participants (46%) reported no change in their dream recall (mean dream recall in the last 7 days = 2.73±1.99) as compared to past year. However, 328 (34%) reported an increase (mean dream recall in the last 7 days = 3.49±2.05; 28% higher than no change group) and 188 (19%) reported a decrease (mean dream recall in the last 7 days = 2.14±2.17; 22% lower than the no change group) in dream recall.

The weekly frequency of bad dreams decreased from the year before the pandemic to the last 7 days ($z = -5.799$, $p < .001$). Mean bad dream recall was 1.56±4.71 bad dreams per week in the year pre-pandemic, and 1.42±1.86 bad dreams per week in the last 7 days. (Median before pandemic = .5, median in the last 7 days = 1 bad dream per week).

## TDQ individual items

**Fig 1. Frequency of 25 most prevalent individual TDQ dream themes.**

The weekly frequency of nightmares, however, did not change (from past year to past 7 days, $z = 1.540$. $p = .123$). Mean nightmare recall was $.82 \pm 2.16$ nightmares per week during the year pre-pandemic, and $.76 \pm 1.52$ nightmares per week during the last 7 days.

There was no significant correlation between sleep duration and dream recall. However, weak correlations were revealed between shorter sleep duration and greater bad dream recall ($r_s = -.116$, $p < .001$), and nightmare recall ($r_s = -.178$, $p < .001$).

**3.1.2. Dream themes and topics during the pandemic.** The following dream themes (individually, from 60 items of the modified TDQ) were the five most prevalent: 1) Trying over and over to do something (51.55%); 2) Sexual experiences (41.32%); 3) Arriving too late (32.95%); 4) Being separated from a loved one (29.65%); 5) Being chased or pursued (29.24%). The four TDQ items that comprised the topic of pandemic imagery, individually, had the following frequencies: 1) Being separated from a loved one (29.65%); 2) Being sick (14.77%); 3) Being in a hospital (12.00%); 4) Germs and contamination (7.64%). The frequencies of 25 most common themes are plotted in Fig 1.

When grouped by larger topics, the prevalence was as follows (% of participants reporting dreaming of a theme at least once in the past 7 days): 1) Inefficacy (62.40%); 2) Human threat (42.46%); 3) Death (38.95%); 4) Pandemic (37.19%); 5) Paralysis (27.79%); 6) Paranormal (26.24%); 7) Natural threat (18.60%). These results are presented in Fig 2.

**3.2.3. Dream recall, dream themes, stress and sleep.** A MANCOVA revealed a statistically significant difference between the stress level groups on the combined dependent variables ($F(6) = 17.698$, $p < .001$; Wilks' $\Lambda = .894$; partial $\eta^2 = .055$). Separate univariate follow-up

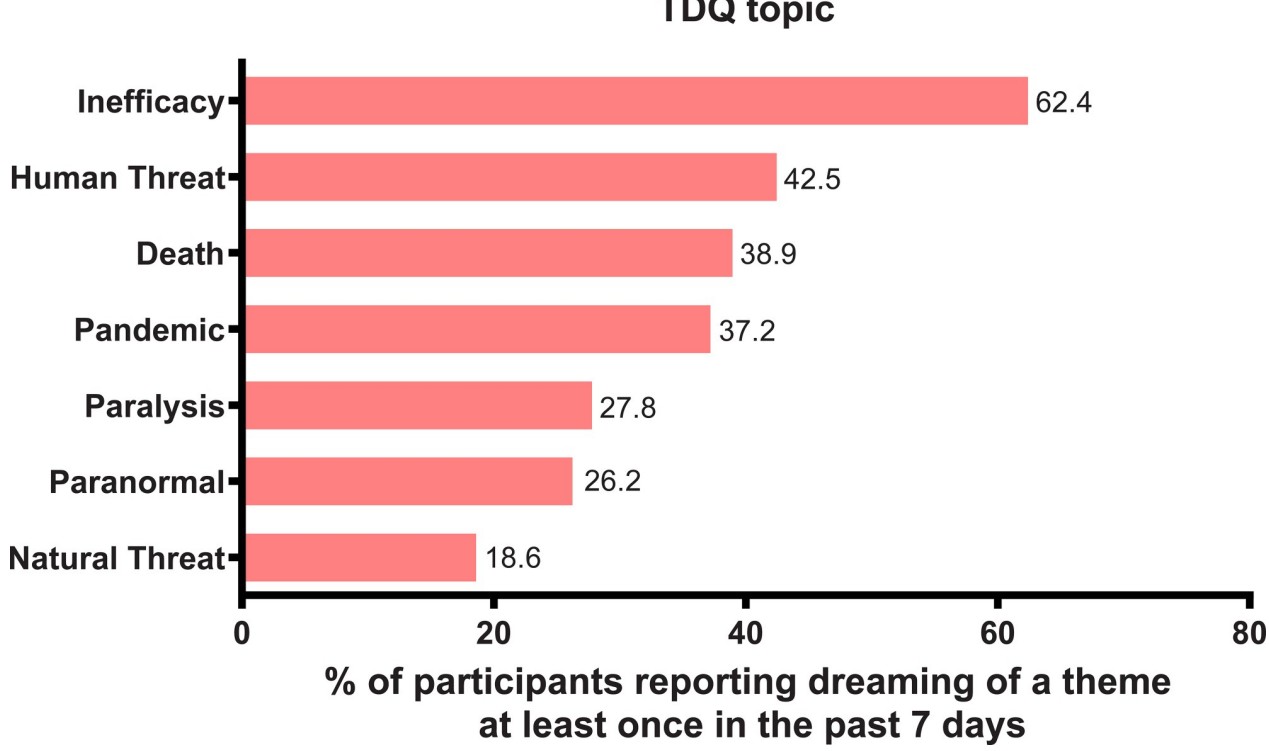

**Fig 2. Frequency of the 7 dream themes.**

ANOVAs with stress levels (low, moderate, severe) as independent factors and dream, bad dream and nightmare recall in the last 7 days, as well as sleep duration, as dependent factors, were performed. A separate ANOVA revealed that sleep duration decreased significantly as stress levels increased, and the frequency of all three dream recall variables increased as stress increased (Table 2).

**Weekly dream recall:** Tukey post-hoc comparisons revealed significant differences between low and severe stress groups (mean difference = .762; $p$ = .003), and medium and severe stress groups (mean difference = .579; $p$ = .038). No difference was observed between low and medium stress groups. **Weekly bad dream recall:** Tukey post-hoc comparisons revealed significant differences between all groups (low < moderate stress: mean difference = .845; moderate < severe stress: mean difference = 1.11; all $p$ < .001). **Weekly nightmare recall:** Tukey post-hoc comparisons revealed significant differences between all groups (low < moderate stress: mean difference = .367; moderate < severe stress: mean difference = 1.12; all p≤.001). **Sleep duration:** Tukey post-hoc comparisons revealed significant differences in hours spent asleep between low and severe stress groups (mean difference = 1.02, $p$ < .001), but not between low and moderate stress groups.

A MANCOVA revealed a statistically significant difference between the stress level groups on the combined dependent dream topic variables ($F(8)$ = 7.021, $p$ < .001; Wilks' $\Lambda$ = .941; partial $\eta^2$ = .030). Further, separate univariate ANCOVAs, with stress levels (low, moderate, severe) as independent factors, weekly dream recall as a covariate, and frequency of dreams associated with the four most prevalent dream topics (inefficacy, human threat, death and pandemic) as dependent measures, were performed (for means, standard deviations, F, $p$, and $\eta_p^2$ values please see Table 2). Frequency of all dream themes significantly increased as stress levels increased (all $p$ < .001).

**Table 2. Comparison of low, moderate and severe stress groups on dream recall, dream topics and sleep measures during the pandemic.**

|  | All participants | Low stress | Moderate stress | Severe stress | F | p | $\eta_p^2$ |
|---|---|---|---|---|---|---|---|
| n | 968 | 443 | 406 | 94 |  |  |  |
| **Weekly dream recall** | (M±SD) | (M±SD) | (M±SD) | (M±SD) |  |  |  |
| Dream recall | 2.9±2.1 | 2.7±2.1 | 2.9±2.0 | 3.5±2.1 | 5.36 | .005 | .01 |
| Bad dream recall | 1.4±1.9 | 0.8±1.4 | 1.7±1.9 | 2.8±2.5 | 57.35 | <001 | .11 |
| Nightmare | 0.8±1.5 | 0.4±1.2 | 0.8±1.3 | 1.9±2.7 | 41.96 | < .001 | .08 |
| **Sleep duration** (hours per day) | 7.3±1.6 | 7.5±0.1 | 7.2±0.1 | 6.4±0.2 | 16.37 | < .001 | .03 |
| **Dream Topics (TDQ)** |  |  |  |  |  |  |  |
| Pandemic | 4.9±1.7 | 4.6±1.2 | 5.0±1.7 | 6.0±2.7 | 26.38 | < .001 | .05 |
| Death | 5.8±1.7 | 5.6±1.3 | 5.9±1.7 | 6.7±2.6 | 13.83 | < .001 | .03 |
| Human Threat | 3.9±1.4 | 3.6±1.1 | 3.9±1.4 | 4.8±2.3 | 27.00 | < .001 | .06 |
| Inefficacy | 6.7±2.2 | 6.3±1.8 | 6.9±2.2 | 7.8±2.9 | 17.84 | < .001 | .04 |

Bonferroni correction: α = .05/8 comparisons = .006.

Results of these ANOVAs and ANCOVAs are also represented in Fig 3.

**3.2.4. Relationship between dream recall, dream themes, stress, anxiety, depression and pandemic concerns.** Spearman correlation analyses between dream, bad dream and nightmare recall and measures of anxiety (GAD7), depression (QIDS) and stress (PSS) revealed positive relationships between all measures of dream recall and of psychopathology. The strongest correlations were weak and were found between frequency of bad dreams and stress, anxiety and depression symptoms. Dream recall was also associated with higher stress and anxiety, although these effects were small. The Spearman *rho* values and significance levels are reported in Table 3.

Spearman correlations between the frequency of the top four dream topics (inefficacy, human threat, death and pandemic imagery) and symptoms of anxiety (GAD7) and depression (QIDS), measures of stress (PSS), and concerns over the pandemic (whether the pandemic was a threat to their health, job, or finances, and specific socio-medical concerns), were performed. All four dream topics were weakly positively correlated with stress, anxiety, and depression. Very weak positive associations were found between some of the frequencies of dream themes and pandemic concerns. For Spearman *rhos* and significance levels please see Table 4.

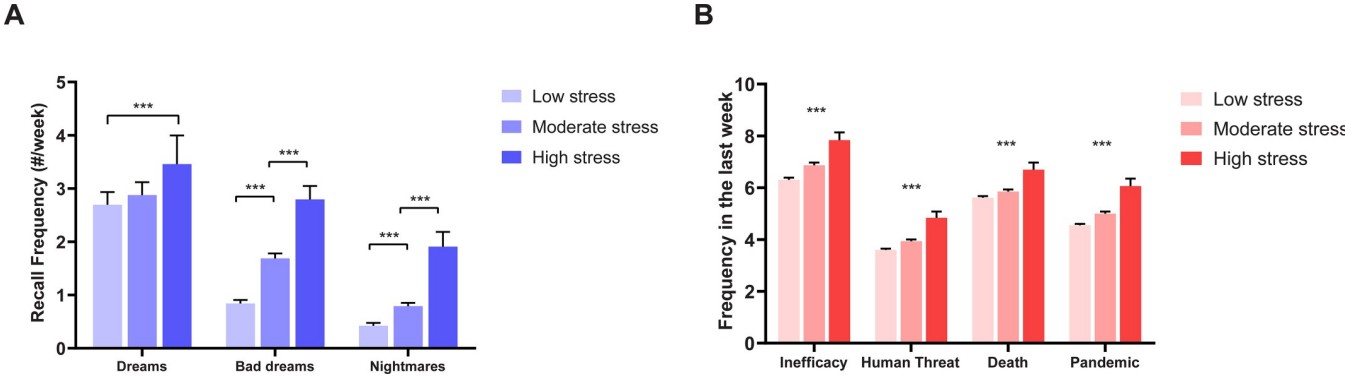

**Fig 3.** Comparison of low, moderate, and high stress groups on (A) dreams, bad dreams, and nightmares recall in the last 7 days, and on (B) the frequency of the top four dream topics (Inefficacy, Human Threat, Death and Pandemic) in the last 7 days. ***p < .001, Bonferroni-corrected for multiple comparisons.

**Table 3. Spearman correlations between dream, bad dream and nightmare recall, and anxiety, depression and stress symptoms.**

|  | PSS (stress) | QIDS (depression) | GAD7 (anxiety) |
|---|---|---|---|
| dream recall | .124* | .155 | .133* |
| bad dreams recall | .363* | .385* | .423* |
| nightmares recall | .100 | .092 | .091 |

*p < .001 Bonferroni correction: α = .05/9 comparisons = .005.

**3.2.5. Linear regression analyses.** *3.2.5.1. Stress levels.* Multiple linear regression model statistically significantly predicted PSS scores $F(12, 748) = 29.248$, $p < .001$, adj. $R^2 = .32$. Among dream variables, nightmare and bad dream frequency (both p < .001) and the pandemic dream topic significantly predicted PSS scores (p = .037). The death topic showed a non-significant trend (p = .075). In addition, younger age (p < .001), female sex (p = .013) and antidepressant use (p = .001) predicted PSS scores.

*3.2.5.2. Depression.* Multiple linear regression model statistically significantly predicted QIDS scores $F(12, 751) = 32.354$, $p < .001$, adj. $R^2 = .33$. Among dream variables, nightmare and bad dream frequency (both p < .001) and death dream topic (p < .001), and inefficacy dream topic (p = .032) significantly predicted QIDS scores. In addition, younger age (p < .001), lower income status (p = .037) and antidepressant use (p < .001) predicted QIDS scores.

*3.2.5.3. Anxiety.* Multiple linear regression model statistically significantly predicted GAD7 scores $F(12, 779) = 32.127$, $p < .001$, adj. $R^2 = .33$. Among dream variables, nightmare and bad dream frequency (both p < .001), dream recall frequency (p = .010), pandemic dream topic (p < .001) and death dream topic (p < .029), significantly predicted GAD7 scores. In addition, younger age (p < .001), female sex (p = .018), and antidepressant use (p < .031) predicted GAD7 scores. These results are presented in Table 5.

## 4. Discussion

During the acute phase of the first COVID-19 lockdown (April 3-May 15, 2020), over half of the respondents to our national longitudinal online survey reported a change in habitual

**Table 4. Spearman correlation coefficients between dream topics and anxiety and depression symptoms, stress, perceptions of COVID-19 threat, and socio-medical concerns associated with COVID-19.**

|  | Inefficacy | Human threat | Pandemic | Death |
|---|---|---|---|---|
| **Mental health** |  |  |  |  |
| GAD7 (anxiety) | .280* | .278* | .304* | .150* |
| QIDS (depression) | .297* | .310* | .266* | .194* |
| PSS (stress) | .234* | .235* | .261* | .125* |
| **COVID-19 Threats** |  |  |  |  |
| health | ns | ns | ns | ns |
| job | ns | ns | ns | ns |
| finances | ns | -113* | ns | -.101 |
| **COVID-19 concerns** |  |  |  |  |
| lacking food | .077 | .142* | .122* | ns |
| public services shutting down | .095 | .147* | .179* | .092 |
| schools closed | .086 | ns | .106 | .086 |
| children or relatives not coping well | .155* | .154* | .164 | ns |
| not being able to access medical services | .147* | .155* | .181* | .135* |

*p < .001 Bonferroni correction: α = .05/44 comparisons = .001.

**Table 5. Multiple regression results for PSS, QIDS and GAD7 scores.**

| | B | 95% CI for B | | SE B | β | $R^2$ | $\Delta R^2$ |
|---|---|---|---|---|---|---|---|
| | | LL | UL | | | | |
| **PSS** | | | | | | | |
| Model | | | | | | .319 | .308*** |
| Constant | 21.499*** | 19.540 | 23.457 | .998 | | | |
| Nightmare frequency | .755*** | .356 | 1.154 | .203 | .138*** | | |
| Bad dream frequency | .592*** | .260 | .923 | .169 | .130*** | | |
| Dream recall | -.214 | .066 | .775 | .142 | -.051 | | |
| Pandemic dream topic | .596* | .036 | 1.156 | .285 | .073* | | |
| Death dream topic | .609 | -.060 | 1.279 | .341 | .067 | | |
| Inefficacy dream topic | .394 | -.113 | .902. | .258 | .055 | | |
| Human threat dream topic | .077 | -.687 | .841 | .389 | .008 | | |
| Age | -.163*** | -.194 | -.131 | .016 | -.333*** | | |
| Sex | -1.442* | -2.575 | -.309 | .577 | -.079* | | |
| Income status | 1.095 | -.498 | 2.668 | .811 | .042 | | |
| Employment status | -.024 | -.076 | .027 | .026 | .138 | | |
| Antidepressant use | 1.954*** | .826 | 3.083 | .575 | .108*** | | |
| | **B** | **95% CI for B** | | **SE B** | **β** | **$R^2$** | **$\Delta R^2$** |
| | | LL | UL | | | | |
| **QIDS** | | | | | | | |
| Model | | | | | | .341 | .330*** |
| Constant | 10.429*** | 9.214 | 11.644 | .619 | | | |
| Nightmare frequency | .537*** | .288 | .786 | .127 | .155*** | | |
| Bad dream frequency | .403*** | .196 | .610 | .105 | .140*** | | |
| Dream recall | -.104 | -.278 | .071 | .089 | -.039 | | |
| Pandemic dream topic | .119 | -.232 | .470 | .179 | .023 | | |
| Death dream topic | .857*** | .440 | 1.275 | .213 | .148*** | | |
| Inefficacy dream topic | .346* | .031 | .662 | .161 | .076* | | |
| Human threat dream topic | .259 | -.216 | .733 | .242 | .040 | | |
| Age | -.074*** | -.093 | -.054 | .010 | -.238*** | | |
| Sex | -.700 | -1.406 | .006 | .360 | -.061 | | |
| Income status | 1.044* | .061 | 2.028 | .501 | .063* | | |
| Employment status | -.020 | -.052 | .012 | .016 | -.036 | | |
| Antidepressant use | 1.692*** | .989 | 2.394 | .358 | .148*** | | |
| | **B** | **95% CI for B** | | **SE B** | **β** | **$R^2$** | **$\Delta R^2$** |
| | | LL | UL | | | | |
| **GAD7** | | | | | | | |
| Model | | | | | | .340 | .329*** |
| Constant | 9.124*** | 7.810 | 10.437 | .669 | | | |
| Nightmare frequency | .686*** | .418 | .957 | .137 | .184*** | | |
| Bad dream frequency | .525*** | .303 | .748 | .113 | .169*** | | |
| Dream recall | -.246** | -.434 | -.058 | .096 | -.087 | | |
| Pandemic dream topic | .683*** | .308 | 1.058 | .191 | .124*** | | |
| Death dream topic | .501* | .051 | .950 | .229 | .080* | | |
| Inefficacy dream topic | .261 | -.079 | .601 | .173 | .053 | | |
| Human threat dream topic | .236 | -.275 | .748 | .261 | .034 | | |
| Age | -.089*** | -.110 | -.068 | .011 | -.267*** | | |
| Sex | -.915* | -1.675 | -.155 | .387 | -.074* | | |

(*Continued*)

**Table 5.** (Continued)

| | | | | | | | |
|---|---|---|---|---|---|---|---|
| Income status | .014 | -1.044 | 1.073 | .539 | .001 | | |
| Employment status | .028 | -.062 | .007 | .018 | -.048 | | |
| Antidepressant use | .835* | .076 | 1.594 | .387 | .068* | | |

Model = "Enter" method in SPSS Statistics; $B$ = unstandardized coefficient; CI = confidence interval; LL = lower limit; UL = upper limit; SE $B$ = standard error of the coefficient; $\beta$ = standardized coefficient; $R^2$ = coefficient of determination; $\Delta R^2$ = adjusted $R^2$.

\*$p < .05$

\*\*$p < .01$

\*\*\*$p < .001$. Sex binary measure: 0 = female, 1 = male.

patterns of dream recall (34% reported increased and 19% reported decreased dream recall). The most prevalent dream themes centered around topics of inefficacy, threat from other humans, death, and pandemic imagery. Pandemic-related dreams were reported by 37% of our participants, while dreams characterized by inefficacy were reported by 62% of participants, with the most prevalent individual theme being trying again and again to do something (51%). The frequency of dream, bad dream and nightmare recall was associated with concomitant levels of stress, with higher frequency of recall in higher stress groups. Similarly, frequency of dreams centered around the four most prevalent dream topics (inefficacy, human threat, death and pandemic imagery) was also associated with higher levels of stress. Lastly, bad dream recall and the four most prevalent dream topics were all also positively correlated with symptoms of anxiety and depression.

## 4.1. Dream, bad dream and nightmare recall

In our sample, the majority of participants reported a change in dream recall during the pandemic lockdown (last 7 days) in comparison to their retrospective estimate (last year). Over a third (34%) of participants reported an increase in dream recall, while almost a fifth of participants (20%) reported a decrease, and about half (46%) showed no change in dream recall. This pattern is somewhat consistent with findings of a survey study by Schredl and Bulkeley [52]: 29% of participants of a US-based survey (n = 3,031) stated that they remembered more dreams than before the pandemic. In contrast, in our sample, we report a higher (19% vs. 7% in Schredl & Bulkeley study) proportion of participants whose dream recall has decreased. This discrepancy is potentially due to differences in methodology: we compared retrospective estimates of dream recall (past year and past 7 days) to determine whether dream recall has changed, while in Schredl & Bulkeley's survey, participants were asked to rate whether their dream recall decreased, stayed the same or increased. Contrary to our expectations, and the existing literature, in our sample, overall bad dream and nightmare recall did not increase during the lockdown as compared to the past year.

Further, and contrary to the sleep extension hypothesis of increased dreaming during the pandemic [72], we did not find an association between dream recall and sleep duration. We did, however, find weak negative correlations between sleep duration and bad dream and nightmare recall, which suggests that shorter sleep time in our sample was associated with higher rates of dysphoric dreaming specifically. This finding is consistent with a recent study in Chinese adolescents showing an association between short sleep time and recurrent nightmares [73]. However, other studies either report a positive relationship between nightmare frequency and longer sleep duration [47, 74] or no association between nightmares and sleep duration [75]. It is possible, nevertheless, that a combination of sleep disturbances during the lockdown, such as shorter sleep duration, has a different relationship to disturbed dreaming

than habitual patterns of shorter or longer sleep time. We previously observed three general patterns of sleep change since the pandemic [20], identifying three clusters: reduced time in bed, delayed sleep and extended time in bed. The two first groups showed higher rates of symptoms of stress, anxiety and depression than the latter group, suggesting that sleep extension potentially had a buffering effect on the experience of lockdown challenges, while reduced sleep time potentially represented maladaptive behavioural and psychological responses to lockdown.

## 4.2. Dream themes

Overall, the most common dream themes corresponded relatively well to existing pre-pandemic literature. However, some changes in the prevalence of themes were also observed during lockdown. Strikingly, the most common dream theme was "trying over and over again to do something", and it significantly surpassed the "being chased or pursued" dream theme which was most commonly reported before the pandemic in the Canadian sample [48], second most prevalent in the German [49] and Chinese [50] samples, after "school, teachers and studying" in both latter cases. In addition, the new theme, "being separated from a loved one" was also highly prevalent. These results suggest that typical dream themes appear to be relatively stable (being chased, sexual experiences, school, falling, arriving late, etc.) despite potentially important changes in organization of individual lives (in this case, during the pandemic-related lockdown). Second, the specific themes that become more pronounced during particular individual and collective challenges, reflect, broadly, the ongoing situation. In our case, items such as "trying" and "being separated from a loved one" were some of the most prominent, representing global psychological concerns during this time.

Dream themes centered around specific elements associated with the pandemic, with the exception of "being separated from a loved one" (that is, hospitals, being sick, germs and contamination) were not very prevalent individually (7.64% for germs/contamination; 12% for being in a hospital; and 14.8% for being sick), and when put together represented the fourth most prevalent topic, after inefficacy, human threat and death imagery. This is consistent with the idea that the dreaming mind, instead of incorporating episodic memories directly and explicitly into oneiric content, extracts the gist of the experience. Indeed, the COVID-19 pandemic (as opposed, for example, to other important individual and collective events), during the first lockdown, was not characterized by particularly recognizable or striking imagery. With the exception of face coverings (which were not mandatory in Canada during the lockdown in April and May 2020), for the majority of respondents, who were not considered essential workers, the pandemic played out mostly outside of the walls of their houses. Considering this, it is perhaps not surprising that the most common dream themes were associated with an overarching topic of inefficacy. Dream themes associated with the topics of inefficacy, human threat, and increased bad dreams and nightmares in medium and high stress groups, also potentially reflected the psychosocial and affective challenges that many faced during the lockdown. Studies from numerous countries report elevated rates of distressing experiences and emotions associated with the emerging pandemic [76]. Fear in general has been reported by multiple teams [77–79]. Among other emotional responses found in the current literature, which may be congruent with the most common dream themes, were anger, hopelessness [78], and loneliness [80].

The fact that pandemic imagery was not reported by the majority of participants, despite its salient nature, is consistent with the rates of 8% of pandemic-related dreams reported in a U.S.-based survey [52], and with the Italian study [65] that reports that 20% of participants had a dream about the pandemic. This finding is also consistent, in a more general way, with studies

of stimuli incorporation into dreams. For instance, a study tracking students' dreams during the examination period [81] did not reveal an increase of dreams about examinations. The mechanisms of stimuli incorporation and functions of memory sources in dreams are still unclear. Indeed, the belief update mechanisms by which (and speed at which) environmentally and culturally novel information can be integrated and automatized remain poorly understood [82].

In fact, while elements of lived experience (memories, thoughts, emotions) make up dream content, full episodic memory incorporations are virtually absent in healthy individuals [35, 83]. Rather, dreams integrate memories in an associative way, bringing together elements of temporally and contextually different episodic memories to create novel scenarios [39, 84]. One of the proposed potential functions of dreams is that the dreaming brain extracts the gist of autobiographically salient experiences [33, 34, 85, 86]. In accordance with this view, we propose that the increase in inefficacy dreams reported in our sample reflected the specific aspects of diminished life opportunities and life plans put on hold under lockdown. Similarly, the high prevalence of dreams of "being separated from a loved one" found in our sample reinforces the notion that dreams preferentially incorporate elements of waking life that are personally significant and reflect our species' inextricably social, relational, and affective nature [87]. Absence of loved ones may not be directly attributed to the pandemic imagery in itself, but social deprivation during lockdown is certainly characterized by a very marked lack of opportunity to spend time with others. Overall, it is likely that environmentally novel information related to the COVID-19 pandemic would not have been sufficiently entrained to be automatized into dream content (the way one would not think in a foreign language through mere exposure while traveling), whereas evolutionarily invariant attentional biases toward threat- and infection-detection, and maintaining social bonds and obligations would flare-up in times of increased uncertainty.

## 4.3. Dreams, nightmares, stress and symptoms of depression and anxiety

Contrary to other studies [53–57], we did not observe an overall increase in bad dream or nightmare frequency during the lockdown, as compared to the pre-lockdown subjective estimate. This is nonetheless consistent with the current research suggesting that propensity towards non-traumatic nightmares may be a relatively stable trait, reflecting a global tendency towards affect distress [88], which potentially develops from early experiences with adversity [30] and alters both the overall affect regulation pathways [89, 90] and cognitive appraisal of the experience of nightmares [91]. Both nightmares [29] and sleep quality in general [92] have been proposed to be representative of stress reactivity. This reactivity may be a trait, which, according to stress-diathesis and differential susceptibility theories [93, 94], predisposes an individual towards a stronger response to environmental stressors. Thus, increased nightmare and bad dream frequency in individuals suffering from moderate to severe stress response during the lockdown may indicate a cross-state reactivity phenomenon.

Our team has previously reported that shorter sleep duration was associated with increased stress levels in the same sample [9]. In the present study we did not find an association between sleep duration and dream recall, with the exception of bad dreams, which suggests that dream recall in general may be independently associated with stress, and thus may represent an indirect marker of stress reactivity/psychological distress.

Further, weak positive associations/trends were found for the relationships between the four most prevalent dream themes (inefficacy, human threat, pandemic and death) and concerns over COVID-19-related social changes (lacking food, public services shutting down, school closures, children or relatives not coping well, and not being able to access medical services).

Lastly, in addition to the well-established notion that dysphoric dreaming reacts to stressors and is affected by psychopathology, an emerging body of research suggests that bad dreams and nightmares may exacerbate or even trigger increased stress response [95] and contribute to daytime distress [91, 96, 97] and symptoms of psychopathology, including anxiety [98] and depression [99]. Thus, to account for the possibility of the bi-directional relationship between dreams, stress and psychopathology, we tested whether dream, bad dream and nightmare frequency, and the most frequent dream topics predicted levels of stress and symptoms of depression and anxiety. Bad dreams and nightmares predicted stress, depression, and anxiety in our sample; pandemic-themed dreams predicted levels of stress; inefficacy dreams predicted symptoms of depression, and death dreams predicted both depression and anxiety symptoms. In addition, female gender predicted stress and anxiety levels, and younger age and antidepressant use predicted all three outcome variables.

Further, different dream topics were associated with different mental health outcomes. Pandemic-themed dream frequency was associated only with stress, suggesting that offline preoccupation with the ongoing health crisis can be seen as a feedback loop factor which is both caused by stress associated with the pandemic and, in turn, may increase stress levels. This is consistent with general theories of nightmares and affect distress [29, 89, 100, 101], lending further support to the notion that a cross-state environmental reactivity or susceptibility trait may serve as a predisposing factor for distress response in the face of a challenging situation.

Inefficacy dreams, the most common dream topic in our sample, predicted only symptoms of depression and not stress or anxiety. The topic of inefficacy is characterized by themes of not being able to do what one needs or wants to and not being able to accomplish one's goals. All these themes can be related to some of the prominent characteristics of depressive symptoms, such as helplessness [102], a tendency towards external locus of control [103], and low self-efficacy [104].

Death dreams, characterized by themes such as seeing someone alive as dead and encountering people who are dead, dying or killing, on the other hand, predicted both depression and anxiety. The topic of death dreams is related to the most salient fears in the context of the pandemic. Death anxiety has been previously associated with depression [105], and more recently, a novel concept of coronaphobia [106] was proposed to describe distressing psychological symptoms, including anxiety and depression, associated specifically with fears related to COVID-19. In a recent study, coronaphobia was positively associated with depression, anxiety and death anxiety [107]. Consistent with literature on the relationship between current preoccupations and dream content, these results indicate an affective cross-state continuity between daytime and nighttime experiences, and suggest a potential role for dreams in emotion regulation and in accentuating symptoms of psychopathology.

### 4.4. Methodological issues and limitations

One of the main limitations of the present study is the self-selected nature of our participants. The TDQ was part of the optional component of the "How are you coping" survey: participants first filled out the main questionnaire and then were asked if they wanted to answer more questions. Thus, our sample consisted of motivated and interested participants, potentially reducing generalizability to the general population. In addition to this selection bias, our sample is potentially characterized by retrospective reporting bias (estimates over the last year vs. the last week). Since we are sampling both time points (before and during the pandemic) at the same time, our study is not truly of longitudinal nature. One of the main strengths of this study, however, is the fact that because the TDQ was part of the optional and additional component of the study, our participants did not specifically expect to be asked questions about

their dreams. In fact, the survey was advertised as a project on mental health and the socio-economic impact of the pandemic. This aspect of our study differentiates our survey from the vast majority of dream research projects, where participants are explicitly recruited to take part in a study on dreams and are potentially self-selected on the basis of their pre-existing interest in dreams. Thus, our sample potentially provides a more generalizable view of oneiric lives during the lockdown.

Further, the pandemic-related themes that we added to the TDQ following the announcement of the first lockdown in March 2020 do not cover the full range of pandemic-related imagery that became salient parts of our lives under the "new normal" as the pandemic progressed (themes such as social distancing, dystopic futures, etc. that were recently identified as common pandemic-related imagery (e.g. [57]) In addition, the TDQ was originally used to identify dream themes over a lifetime, and we assessed them over a much more limited span of the last 7 days in order to sample the most recent dream themes occurring specifically in the context of the lockdown. Therefore, the comparisons with the previous literature have to be nuanced, bearing in mind the temporal scale differences between our study and earlier research.

## 5. Conclusions

We report changes in dream content during the first COVID-19 lockdown (April-May 2020) and a relationship between dreams, bad dreams, nightmares, and dream themes to stress, depression and anxiety. Dream recall, bad dream and nightmare frequency increased during the lockdown, particularly for individuals with moderate to high stress, potentially representing individual tendency towards environmental susceptibility and global traits for affect distress and stress reactivity. The most prevalent dream themes were centered around topics of inefficacy, human threat, pandemic, and death imagery. The frequency of these dreams was positively associated with stress levels, anxiety and depression symptomatology. Lastly, bad dreams and nightmares were associated with symptoms of stress, depression and anxiety, and different dream topics predicted different mental health outcomes. Specifically, pandemic dreams were associated with stress, inefficacy dreams were associated with depression, and death dreams were associated with both depression and anxiety. We suggest that the unique lived experience of life under a pandemic lockdown produced changes not only in general mental health and sleep patterns, but also in oneiric life, reflecting in a general sense, the ongoing existential challenges. Specifically, the most common dream themes during this time centered around the topic of inefficacy (trying over and over again, arriving too late, etc.), and dreams of being separated from loved ones evoke the overall feeling of lives put on hold, and possibly represent the process of making sense of this new experience. These results are consistent with theories emphasizing the associative nature of memory processing in dreams and the role of emotionally charged and personally significant events in dream formation.

## Acknowledgments

The authors are grateful to our participants for taking the time to fill out a battery of questionnaires during this difficult time.

## Author Contributions

**Conceptualization:** Elizaveta Solomonova, Marie-Hélène Pennestri, Mysa Saad, Tetyana Kendzerska, Samuel Paul Louis Veissiere, Roger Godbout, Jodi D. Edwards, Lena Quilty, Rebecca Robillard.

**Data curation:** Mysa Saad.

**Formal analysis:** Elizaveta Solomonova, Rebecca Robillard.

**Investigation:** Elizaveta Solomonova, Rebecca Robillard.

**Methodology:** Elizaveta Solomonova, Marie-Hélène Pennestri, Tetyana Kendzerska, Samuel Paul Louis Veissiere, Jodi D. Edwards, Lena Quilty, Rebecca Robillard.

**Project administration:** Mysa Saad, Tetyana Kendzerska, Rebecca Robillard.

**Resources:** Rebecca Robillard.

**Software:** Rebecca Robillard.

**Supervision:** Elizaveta Solomonova, Rebecca Robillard.

**Writing – original draft:** Elizaveta Solomonova, Claudia Picard-Deland.

**Writing – review & editing:** Elizaveta Solomonova, Claudia Picard-Deland, Iris L. Rapoport, Marie-Hélène Pennestri, Tetyana Kendzerska, Samuel Paul Louis Veissiere, Roger Godbout, Jodi D. Edwards, Lena Quilty, Rebecca Robillard.

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
