## [Decision Letter · Decision Letter 0]

29 Jun 2021

PONE-D-21-16452

Stuck in a lockdown: dreams, bad dreams, nightmares, and their relationship to stress, depression and anxiety during the COVID-19 pandemic.

PLOS ONE

Dear Dr. Solomonova,

Thank you for submitting your manuscript to PLOS ONE. After careful consideration, we feel that it has merit but does not fully meet PLOS ONE’s publication criteria as it currently stands. Therefore, we invite you to submit a revised version of the manuscript that addresses the points raised during the review process.

Although reviewers recommend publication, reviewer 2 suggests some minor revisions to your paper that will improve it further. 

We look forward to receiving your revised manuscript.

Kind regards,

Serena Scarpelli

Academic Editor

PLOS ONE

Reviewers' comments:

Reviewer's Responses to Questions

**Comments to the Author**

1. Is the manuscript technically sound, and do the data support the conclusions?

Reviewer #1: Yes

Reviewer #2: Yes

2. Has the statistical analysis been performed appropriately and rigorously? 

Reviewer #1: Yes

Reviewer #2: Yes

3. Have the authors made all data underlying the findings in their manuscript fully available?

Reviewer #1: Yes

Reviewer #2: No

4. Is the manuscript presented in an intelligible fashion and written in standard English?

Reviewer #1: Yes

Reviewer #2: Yes

5. Review Comments to the Author

Reviewer #1: This is a very well structured and written manuscript on a theme that has already drawn a considerable research attention. The results echo previous studies for the key questions. In this sense, the contribution of this study to the previous studies is being confirmative rather than being entirely novel. Yet, the associations between mental health and nighmares/bad dreams were very weak.

The retrospective estimate of dreaming pre-pandemicly is questionnable though as a comparison variable - therefore the result of no increase of bad dreaming is not very solid.

My impression of the manuscript then is that it is very well constructed - methods, results. discussion and limitations - all are well in place. But the scientifiv contribution of this study is not such strong - negative or null findings may also depend on the context where the survey was conducted, including retrospective evaluations that are not very reliable.

Reviewer #2: The paper is well written. The introduction points to a very recent literature review in an organized sequence, linking with previous literature on dream research.

The methodology is appropriate for the hypothesis and research question and well explained. Results with figures and tables are well described, in a logical sequence of analysis, with rigorous statistics and adequately corrected by multiple comparisons.

At the discussion session, during the discussion of multilinear models, the interpretation suggests causality (“These results indicate a potential role that bad dreams and nightmares may play in amplifying the experience of stress, depression, and anxiety”). Still, the method doesn’t support this conclusion, as it models an association in a single time point. Thus, for example, the association between antidepressants and levels of depression, anxiety, and stress is expected and makes sense (those that in this single time point were expressing higher levels of symptoms were also already in treatment). Still, we cannot say that antidepressants generate higher levels of stress, depression, and anxiety.

The association with mental suffering and antidepressants also flagged one caution related to the effects of antidepressants (or even mental illness diagnosis) into a dream experience. It would be interesting to repeat the analysis on two groups: 1. a group without mental illness diagnosis and no use of psychiatric medication; and 2. a group under psychiatric treatment. Maybe the associations with dreams and psychopathology may reveal different associations for those already under treatment and those that were not.

6. PLOS authors have the option to publish the peer review history of their article (what does this mean?). If published, this will include your full peer review and any attached files.

Reviewer #1: No

Reviewer #2: No

---

## [Author Response · Author response to Decision Letter 0]

13 Sep 2021

We thank the reviewers for their comments and sugesstions. Please find responses to reviewers’ comments below 

Reviewer #1: This is a very well structured and written manuscript on a theme that has already drawn a considerable research attention. The results echo previous studies for the key questions. In this sense, the contribution of this study to the previous studies is being confirmative rather than being entirely novel. Yet, the associations between mental health and nighmares/bad dreams were very weak.

The retrospective estimate of dreaming pre-pandemicly is questionnable though as a comparison variable - therefore the result of no increase of bad dreaming is not very solid.

My impression of the manuscript then is that it is very well constructed - methods, results. discussion and limitations - all are well in place. But the scientifiv contribution of this study is not such strong - negative or null findings may also depend on the context where the survey was conducted, including retrospective evaluations that are not very reliable.

Reviewer #2: The paper is well written. The introduction points to a very recent literature review in an organized sequence, linking with previous literature on dream research.

The methodology is appropriate for the hypothesis and research question and well explained. Results with figures and tables are well described, in a logical sequence of analysis, with rigorous statistics and adequately corrected by multiple comparisons.

At the discussion session, during the discussion of multilinear models, the interpretation suggests causality (“These results indicate a potential role that bad dreams and nightmares may play in amplifying the experience of stress, depression, and anxiety”). Still, the method doesn’t support this conclusion, as it models an association in a single time point. Thus, for example, the association between antidepressants and levels of depression, anxiety, and stress is expected and makes sense (those that in this single time point were expressing higher levels of symptoms were also already in treatment). Still, we cannot say that antidepressants generate higher levels of stress, depression, and anxiety.

RESPONSE: We agree and we deleted that sentence.

The association with mental suffering and antidepressants also flagged one caution related to the effects of antidepressants (or even mental illness diagnosis) into a dream experience. It would be interesting to repeat the analysis on two groups: 1. a group without mental illness diagnosis and no use of psychiatric medication; and 2. a group under psychiatric treatment. Maybe the associations with dreams and psychopathology may reveal different associations for those already under treatment and those that were not.

RESPONSE: Thank you for this suggestion. We have re-done the analyses with the two groups, as the reviewer suggested. Similar associations emerged: for the antidepressant use group, stress was predicted by younger age and nightmares; depressive symptoms were predicted by younger age, nightmares, inefficacy and death dreams; and anxiety was predicted by younger age, nightmares and bad dreams. For the no antidepressant use group, stress was predicted by younger age, female gender, nightmares, bad dreams, pandemic dreams; depressive symptoms were predicted by younger age, female gender, nightmares, bad dreams, pandemic dreams and death dreams; and anxiety was predicted by younger age, female gender, nightmares, bad dreams and pandemic dreams. Considering these very small differneces and overall similar results of the regression analyses for the two groups, we have decided not to include these in the manuscript.

---

## [Decision Letter · Decision Letter 1]

12 Oct 2021

Stuck in a lockdown: dreams, bad dreams, nightmares, and their relationship to stress, depression and anxiety during the COVID-19 pandemic.

PONE-D-21-16452R1

Dear Dr. Solomonova,

We’re pleased to inform you that your manuscript has been judged scientifically suitable for publication and will be formally accepted for publication once it meets all outstanding technical requirements.

Kind regards,

Serena Scarpelli

Academic Editor

PLOS ONE

Additional Editor Comments (optional):

Reviewers' comments:

Reviewer's Responses to Questions

**Comments to the Author**

1. If the authors have adequately addressed your comments raised in a previous round of review and you feel that this manuscript is now acceptable for publication, you may indicate that here to bypass the “Comments to the Author” section, enter your conflict of interest statement in the “Confidential to Editor” section, and submit your "Accept" recommendation.

Reviewer #2: All comments have been addressed

2. Is the manuscript technically sound, and do the data support the conclusions?

Reviewer #2: Yes

3. Has the statistical analysis been performed appropriately and rigorously? 

Reviewer #2: Yes

4. Have the authors made all data underlying the findings in their manuscript fully available?

Reviewer #2: No

5. Is the manuscript presented in an intelligible fashion and written in standard English?

Reviewer #2: Yes

6. Review Comments to the Author

Reviewer #2: (No Response)

7. PLOS authors have the option to publish the peer review history of their article (what does this mean?). If published, this will include your full peer review and any attached files.

Reviewer #2: No

---

## [Editor Report · Acceptance letter]

29 Oct 2021

PONE-D-21-16452R1 

Stuck in a lockdown: dreams, bad dreams, nightmares, and their relationship to stress, depression and anxiety during the COVID-19 pandemic. 

Dear Dr. Solomonova:

I'm pleased to inform you that your manuscript has been deemed suitable for publication in PLOS ONE. Congratulations! Your manuscript is now with our production department. 

Kind regards, 

on behalf of

Dr. Serena Scarpelli 

Academic Editor

PLOS ONE